# From Fruit to Beverage: Investigating *Actinidia* Species for Characteristics and Potential in Alcoholic Drink Production

**DOI:** 10.3390/foods13152380

**Published:** 2024-07-27

**Authors:** Alessandra Di Canito, Alessio Altomare, Nicole Giuggioli, Roberto Foschino, Daniela Fracassetti, Ileana Vigentini

**Affiliations:** 1Department of Biomedical, Surgical and Dental Sciences (DISBIOC), University of Milan, 20122 Milan, Italy; alessandra.dicanito@unimi.it (A.D.C.); roberto.foschino@unimi.it (R.F.); ileana.vigentini@unimi.it (I.V.); 2Department of Food, Environmental and Nutritional Sciences (DeFENS), University of Milan, 20133 Milan, Italy; alessio.altomare@unimi.it; 3Department of Agricultural, Forest and Food Sciences (DISAFA), University of Turin, 10095 Turin, Italy; nicole.giuggioli@unito.it

**Keywords:** *Actinidia*, SunGold, Tahi, Rua, *Torulaspora delbrueckii*, characterization, fruit, juice, aroma, sustainability

## Abstract

There is a growing interest in various types of kiwifruits, such as the “yellow” kiwifruit (*Actinidia chinensis* var. *chinensis*) and the “baby” kiwifruit of *Actinidia arguta*. These fruits are rich in bioactive compounds, which contribute to their nutraceutical properties, but they nevertheless have a shorter shelf life, resulting in economic losses. This study aims to chemically characterize kiwifruit juices from SunGold and baby kiwifruit varieties (Hortgem Rua and Hortgem Tahi) to improve knowledge and explore the suitability of these products for producing low-alcohol beverages using non-*Saccharomyces* strains, with the purpose of reducing waste and generating value-added processing. Total soluble solids, acidity, hardness, dry matter, total phenolic content, and antioxidant capacity were used as indicators of fruit quality. Chemical characterization of fresh kiwifruit juices revealed distinct profiles among varieties, with SunGold juice exhibiting higher sugar content and acidity. Citric acid was the predominant organic acid, while, as expected, tartaric was not detected. Kiwifruit juice fermentations by *T. delbrueckii* UMY196 were always completed regardless of the type of juice used, producing beverages with ethanol content ranging from 6.46 to 8.85% (*v*/*v*). The analysis of volatile organic compounds highlighted the presence of several molecules, contributing to aroma profiles with relevant differences among the three kiwifruit-based drinks. In particular, the total concentration of esters in the beverages reached 8.2 mg/L, 9.2 mg/L, and 8.6 mg/L in the Tahi, Rua, and SunGold beverages, respectively. The qualitative and quantitative profiles of the kiwifruit drinks revealed a pronounced perception of acidity and fruity traits, with significant differences observed by Principal Component Analysis (PCA) in aroma descriptors among the proposed beverages. The possibility of using unsold, overripe, or waste fruits to prepare new products with an increase in the sustainability of this supply chain is set.

## 1. Introduction

Kiwifruit production in Europe concerns about 15% of the world market. Besides the well-known green kiwi (*Actinidia deliciosa*), production and consumer demand regarding different varieties of kiwifruits have increased in the last few years. In particular, interest in the SunGold or “yellow” kiwi (*Actinidia chinensis* var. *chinensis* ‘Zesy003’) and the kiwi belonging to the *Actinidia arguta* species, also known as the baby kiwi or mini kiwi, has grown [1].

The SunGold kiwifruit is native to New Zealand and was developed by the Zespri company. It was later imported to Italy, which has become the world’s main producer of this product, despite it representing only 5% of Italian production. The yellow-fleshed kiwi plant is a climbing plant that can reach up to 10 m in height. In Italy, it is harvested in late autumn. The plant is very similar to the green kiwifruit in terms of cultivation type and seasonality. In addition to the characteristic golden yellow color, the SunGold kiwi differs from the classic green kiwifruit in flavor, being much sweeter, and with smoother skin [2]. Another difference is the price, which is almost double that of the classic green kiwifruit. In terms of nutritional composition, SunGold kiwifruit is a product rich in water, with few carbohydrates (15.8%), of which sugars represent 12.3%, providing only 63 kcal per 100 g of product [3]. In addition, it is an excellent source of fiber (1.4%), potassium, vitamin C, folic acid, and vitamin E. Due to its yellow color, it is rich in carotenoids and flavonoids. On the other hand, baby kiwis are becoming known for their interesting nutraceutical properties, such as a high content of bioactive compounds and vitamin C [4,5], as well as a sweet and aromatic taste, contributing to their categorization as healthy fruits [6]. These fruits are derived from different geographical origins [7,8], and the Hortgem Tahi^®^ and Hortgem Rua^®^ cultivars are the most developed in Italy, marketed under the Nergi^®^ brand [2].

The Tahi kiwifruit is native to Asia, where it grows as a climbing plant. It is cultivated worldwide in countries such as New Zealand, Chile, Spain, Portugal, and many others. The species adapts well to slightly acidic, well-drained, and loose soils. In the past, it was a wild plant and not cultivable because the available plants produced very vulnerable fruits that were difficult to preserve and maintain. Only in the 1990s was the problem solved, when some plants of this species were crossed with each other in order to improve their overall quality. This led to the emergence of new varieties, which maintained the sensory characteristics and nutritional qualities of the original varieties, differing only in better manageability and greater fruit preservation. In an orchard, the plant can reach up to 6 m in height, with flowering in May and fruit production in mid-summer. A single plant can produce from 30 to 50 kg of fruits, which are harvested solely by hand in the period between late August and early September. The baby kiwifruit is characterized by its small size (2–3 cm) and by a green and edible peel that gives an intense and sweet taste. Additionally, it contains large quantities of vitamin C (52.5 mg/100 g product) and vitamin E (5.28 mg/100 g product), and it is also rich in calcium, magnesium, phosphorus, and fiber [3,9].

The Rua variety grows on a plant known as “Ken’s Red” or “Rua”, which is a climbing fruit plant developed in New Zealand that can reach significant dimensions. It produces small-sized fruits with thin red-violet skin, tender red pulp, a very sweet taste, and rich in vitamins C, B, and E. The fruits mature from late September onwards. Despite having a very low energy capacity (providing 52 kcal per gram), they have excellent nutritional characteristics [10].

Globally, the discarded fruit in the production phase and along the supply chain represents 10^6^ tons of annual waste [5]. The fruit of *Actinidia arguta* has a short shelf life (1–2 months) compared to the green kiwifruit (6–8 months) and the yellow kiwifruit (4–6 months) [11,12,13], and their quality during the harvest influences the storage process [14]. Furthermore, the rapid deterioration of baby kiwis results in significant economic and environmental losses [15,16], thus underscoring the necessity for enhanced production and distribution management strategies. Fracassetti and co-authors [17] proposed the possibility of using kiwifruit juice in a co-fermentation process to obtain kiwi-based wines in order to exploit the surplus or the waste derived from kiwifruit production; this process could limit post-harvest loss and waste. However, knowledge of the chemical characterization of alcoholic beverages from kiwi fruits is limited [18]. Even if the chemical and sensory characteristics of kiwi wine are related to the fermentation process, the sugar/acid ratio of raw material at ripening is fundamental for the production of a balanced kiwi wine. Kiwifruit wine processed with pectinase is reported to have superior sensory value, ethanolic strength, and total antioxidant activity if made from overripe berries [19].

The aim of this study was the chemical characterization of kiwi juices from *Actinidia chinensis* var. *chinensis* ‘Zesy003‘(SunGold) and *Actinidia arguta* cv. Rua and Tahi (baby kiwi) in order to expand the knowledge related to these fruits being relevant for the agri-food field. The suitability of juices from these kiwifruit species for the production of low-alcoholic beverages using a non-*Saccharomyces* strain was investigated. The outcome of this work may facilitate the reduction of waste and the market expansion of these kiwifruit species through the implementation of alternative processing methods.

## 2. Materials and Methods

### 2.1. Fruit Material and Preparation of Juices

The fruit juices were obtained from *Actinidia chinensis* var. *chinensis* ‘Zesy003’ (SunGold) and *Actinidia arguta* cv. Rua and Tahi. All fruits had Italian origin and were collected at the commercial ripening phase by purchasing them from a large-scale retail store. Yellow and baby kiwis were purchased, respectively, from the Zespri^®^ (Berchem, Belgium) and Nergi^®^ (Labatout, France) brands. The berries were not subjected to treatment prior to analysis. After preliminarily peeling the yellow kiwis, the fruits of each cultivar were cut and squeezed with a juice extractor, obtaining a juice extraction yield of up to 70%, specifically 70%, 32%, and 65% for SunGold, Rua, and Tahi, respectively. The resulting juices were added to potassium metabisulfite (50 mg/L) and stored at −18 ± 1 °C before characterization analysis.

### 2.2. Chemical and Nutraceutical Evaluation of Kiwifruits

Twelve samples per variety were used in the physiochemical analysis. Total soluble solids (TSS/°Brix) were determined using a digital refractometer (Atago, PAL-1). Pulp fruit from each sample was extracted and the titratable acidity (TA) was measured with an automatic titrator; results were expressed as % (*w*/*v*) of citric acid. Sample hardness (N) was quantified using a texture profile analyzer (TPA) with the use of a 5 mm standard penetration head. The TPA was set at 5 g trigger force, 1 mm/s measure speed, 2 mm measure distance, and 1 mm/s reverse and forward speed. The data were recorded using FTA standard software 3.0. Dry matter (DM) was estimated by drying three replicates of approximately 20 g of material in an oven at 70 °C (overnight). The fresh and dry weight data were used to calculate the corresponding dry matter percentage. For total polyphenolic compounds and antioxidant capacity, an extraction solvent (12.5 mL) was added to 5 g of fruit material. The extraction solvent was prepared fresh prior to the analysis (500 mL methanol, 23.8 mL water, and 1.4 mL 37% *w/v* HCl). Samples were incubated for 1 h at room temperature in the dark. Next, samples were homogenized for 1 min followed by centrifugation for 15 min at 4500× *g* (Rotofix 32, Hettich Zentrifugen, Tuttlingen, Germany). Total polyphenol index (TPI) and antioxidant capacity were analyzed following the Slinkard and Singleton (1977) protocol using the same extraction solution [20]. TPI was determined as follows: 250 μL of the sample in the extracted solution was added to 18.5 mL of water (75 dilution), 1.25 mL of Folin–Ciocalteu reagent (Merck, Darmstadt, Germania), and 5 mL of 15% (*w*/*v*) sodium carbonate. Absorbance was recorded at 765 nm after 2 h incubation at room temperature. Antioxidant capacity was measured via the FRAP test method. All reagents were prepared fresh prior to the experiment according to the Slinkard and Singleton (1977) [20] protocol. In brief, 900 μL FRAP reagent (25 mL 0.3 M pH 3.6 acetate buffer, 2.5 mL 10 mM 2, 4, 6-tripiridil-s-triazin in 40 mM HCl, 2.5 mL of FeCl_3_∙6 H_2_O; incubated for 15 min in a 37 °C water bath prior to use) was added to 90 μL water and 30 μL sample extraction solution. Samples were incubated for 15 min in a 37 °C water bath. Absorbance was measured at 595 nm.

### 2.3. Chemical Characterization of Kiwifruit Juices

Juices from each kiwifruit variety obtained with the extractor were characterized in terms of titratable acidity (TA) measured with an automatic titrator; results were expressed as % (*w*/*v*) citric acid.

Glucose and fructose were assessed through enzymatic determination with an iCUBIO i-Magic M9 automatic enzymatic analyzer (r-Biopharm, Darmstadt, Germany) according to the instructions of the manufacturer. The enzymatic kits are specific for the investigated substances; no interferences have been observed during the experiments. When necessary, appropriate dilutions of the samples were prepared in order to obtain a linear measurement of the tested compounds. The organic acids were quantified as described by Rustioni and co-authors (2023) [21]. Briefly, an Acquity HClass UPLC (Waters, Milford, MA, USA) system equipped with a photodiode array detector 2996 (Waters) was used. Chromatographic separations were performed with a Synergy, 4 μm HYDRO-RP, 80 A, 250 4.6 mm (Phenomenex, Torrance, CA, USA). The separation was carried out in isocratic conditions using phosphate buffer 20 mM at pH 2.9 at a flow rate of 0.6 mL/min, and the column temperature was 30 °C. Calibration curves were obtained for tartaric, malic, lactic, citric, acetic, and succinic acids at concentrations of 0.1–10 g/L, giving a linear response in the concentration range. Quantification was performed according to the external standard method. Data acquisition and processing were carried out with Empower 2 software (Waters) at 210 nm.

### 2.4. Yeast Strain and Fermentation Trials

Pure cultures of *Torulaspora delbrueckii* UMY196 were used for the fermentation trials. This strain is part of the yeast culture collection of the University of Milan (Italy), and it was isolated from wine. Cells were maintained at −80 °C in YPD medium (10 g/L yeast extract, 20 g/L peptone, 20 g/L glucose, pH 5.5) supplemented with 20% (*v*/*v*) glycerol. Cell pre-cultures were obtained by inoculating 1% (*v*/*v*) glycerol stock freeze culture in YPD broth maintained at 30 °C for 24–48 h in aerobiosis. The final biomass was determined by optical density (OD) at 660 nm. For the inoculum, a volume corresponding to 5 × 10^6^ CFU/mL from the pre-culture was centrifuged at 5000× *g* for 15 min (Hettich, ROTINA 380R, Tuttlingen, Germany), collected, and washed once with 0.9% (*w*/*v*) NaCl. The fruit juices were inoculated at 0.1 ± 0.05 OD_660 nm_, corresponding to about 1 × 10^6^ CFU/mL [17].

Fermentation trials were performed in triplicate under limited oxygen conditions at 25 ± 1 °C in 300 mL flasks containing 200 mL of kiwifruit juice. If necessary, ammonium sulfate was added prior to the yeast inoculum, adjusting the readily assimilable nitrogen (RAN) content to 200 mg/L. The alcoholic fermentation (AF) trend was followed considering the daily weight loss of the inoculated flasks; the fermentation was considered complete when no weight change was observed. At the end of AF, the drinks were centrifuged at 5000× *g* for 20 min at 10 °C (Beckman, CA, USA) to remove the yeast cells. The supernatants were added to potassium metabisulfite (50 mg/L) and stored in capped bottles at 4 ± 1 °C for about one month until the analysis.

### 2.5. Microbial and Chemical Analysis of Kiwifruit Drinks

In this study, the samples underwent standardized conditions and microbiological analysis to ensure controlled fermentation. Colony-forming unit (CFU/mL) enumeration was obtained for yeasts after the inoculum and at the end of the fermentation. The samples were spread on Wallerstein Laboratory (WL) nutrient agar medium (Scharlau, Sentmenat, Spain) in 100 µL decimal serial dilutions, after 2–4 days of incubation at 25 °C in aerobiosis. In addition, after metabisulfite was added overnight, the juices were evaluated for potential contamination.

In addition, molecular identification via sequencing of the internal transcribed spacers between the 18S and 26S rDNA genes (ITS1–5.8S–ITS2) was conducted at both the beginning and the end of the fermentation to confirm the exclusive presence of *T. delbrueckii* UMY196.

Yeast identification was carried out via colony-PCR, with DNA extraction performed as follows: a single full-sized colony was dissolved in 200 μL of ddH_2_O with 20 μL of Zymolase lysis buffer [1 μL of 5 U/μL Zymolase (Zymo Research, CA, USA), 99 μL phosphate buffer] at 37 °C for 2 to 3 h. Subsequently, after heat treatment at 95 °C for 15 min and centrifugation at 2500× *g* for 7 min (Hettich Zentrifugen, Mikro 200, Tuttlingen, Germany), 3 to 5 μL of supernatant were utilized for amplification using primers ITSY1 (5′ TCCG-TAGGTGAACCTGCGG 3′) and ITSY4 (5′ TCCTCCGCTTATTGATATGC 3′) [22].

The PCR mixture comprised 1X Q5 DNA Polymerase buffer (New England, Biolabs, Ipswich, MA, USA), 200 μM dNTPs (New England, Biolabs, Ipswich, MA, USA), 0.5 μM of each primer, 0.02 U/µL Q5 Hot Start High-Fidelity DNA Polymerase (New England, Biolabs, Ipswich, MA, USA), and approximately 100 ng of DNA. The reaction was conducted in Mastercycler nexus (Eppendorf, Hamburg, Germany), with the amplification proceeding as follows: initial denaturation at 98 °C for 5 min, followed by 35 cycles at 98 °C for 10 s, annealing at 50 °C for 30 s, and extension at 72 °C for 1 min. This was followed by a final extension step at 72 °C for 2 min.

Amplification products were resolved via agarose gel electrophoresis in 1.0% (*w*/*v*) agarose gels in TAE buffer (40 mM Tris-acetate, pH 8.2; 1 mM EDTA) at 100 V for 1 h. Gels were stained with 0.5 μg/mL ethidium bromide and photographed under UV illumination (GelDoc XR, BioRad, Hercules, CA, USA). A 100 bp DNA ladder marker (Sharpmass^TM^100, Euroclone SpA, Pero, Italy) served as the size standard.

Subsequently, the amplified products were forwarded for sequencing by an external provider (Eurofins, Milan, Italy). The obtained sequences underwent identification using the BLAST algorithm, which involved comparison with sequences listed in databases (www.ncbi.nlm.nih.gov, accessed on 14 February 2024).

At the end of alcoholic fermentation, the quantification of residual sugars and organic acids was carried out as described in Section 2.3. Moreover, glycerol was determined by the automatic enzymatic analyzer iCUBIO i-Magic M9 (R-Biopharm). The organic acids were quantified as described in Section 2.3, while ethanol production was determined using the r-Biopharm enzymatic kit (r-Biopharm, Darmstadt, Germany). The free and glycoconjugate aromas of kiwifruit juices and the resulting drinks were determined by GC-MS, as described by Mateo et al. [23] and Petrozziello et al. [24]. Based on the known perception thresholds, the odor activity values (OAVs) were calculated as the ratio between the aroma concentration and its perception threshold.

### 2.6. Sensory Analysis

Sensory analysis was carried out for the obtained kiwi drinks. A panel composed of 10 experienced judges (five females, five males, average age 29) considered the following descriptors: peach, floral, passion fruit, and honey for olfactory perception and citrus, fruity, and apple for aftertaste perception. Sweetness, acidity, bitterness, flavor intensity, sapidity, and aftertaste persistency were also evaluated [17]. The descriptors were scored on a nine-point scale, with nine being the highest intensity. The discriminant capacity of the judges was set at 20% and the replicability was set at 75% [25].

### 2.7. Statistical Analysis

One-way ANOVA was determined using SPSS Win 12.0 program (SPSS Inc., Chicago, IL, USA). Differences were evaluated by the post-hoc Fisher LST and the significances were set at a value of *p* < 0.05. The equations of the calibration curves were assessed by linear regression analysis. Principal component analysis (PCA) was performed with Statistica 12 software (Statsoft Inc., Tulsa, OK, USA) on auto-scaled data for an overall overview of the aroma compounds and sensory descriptors.

## 3. Results and Discussion

### 3.1. Physicochemical Characterization of Fresh Kiwifruits

Physicochemical and nutraceutical characteristics were evaluated for the fresh kiwifruit varieties (Table 1). The value of total soluble solids (TSS) is an index of the substances, mainly sugars together with organic acids, minerals, vitamins, and starch, which are accumulated by the fruit during ripening [26], and it is essential as an indicator of the sensory quality of the fruit [27]. Among varieties, the SunGold is sweeter and has the lowest acidity (%) value, while the cv Rua is the baby kiwi with the lowest TSS content. Starch degradation mainly occurs after the fruit is ripe and by that time, almost all of the starch has been converted into soluble sugars. This process is complex and dependent on the osmotic pressure, mainly driven by the present organic acids [28]. The results highlight a significant variability which can be justified by the fact that every single fruit gathers sugars and compounds differently, according to its position in the field and on the tree, as well as the climatic conditions, the production area, and the harvest time. In addition, the ripening stage also affects the sugar level of the berries [29]. With regard to the dry matter (DM) measurements, minimal differences were observed in the baby kiwi samples, whereas variations were evident in the SunGold samples, which had the highest level (22%). DM is highly correlated to the sugar and starch content, and it is of commercial interest, as farmers in New Zealand are paid premium prices for fruit with higher dry matter percentages [30]. This is a clue to use in hardy kiwifruit research, showing that starch accumulation in the kiwifruit family is essential. In particular, Oh et al. [14] indicate that for the hardy kiwifruit, the development of DM, starch, and sugars occurs independently of environmental conditions pre-harvest. In our case, all Nergi^®^ brand samples are from the same pre-harvest cultivation area (Cuneo province, Italy). The levels of acidity suggest that the fruits were ripe, even in the case of the SunGold variety, showing the lowest acidity (Table 1). Considering the development stage of the kiwifruits, the results were in agreement with the literature, where the fruit’s acidity was attributed to osmotic pressure regulation [28,31]. In general, the main acids known to be present in kiwifruits are citric, malic, and quinic acids in concentrations that are strictly developmental stage- and cultivar-dependent [27]. Hardness is an important quality parameter determining the time for consumption of fresh kiwifruit and it is an indicator of storage period management [32]. In the case of SunGold, the two components of the berries should be distinguished as the pulp and the central fibrous core, which is white and firmer because of its composition. The variability of firmness between the pulp and core is dependent on the effectiveness of storage methods in terms of temperature, relative humidity, and storage atmosphere, and it could affect the consumers’ choice as they can decide whether to buy the fruits according to personal preferences. Hardness values found in this work were generally higher for all the kiwifruit varieties considered than those found as outcomes [26]. In any case, the SunGold fruits showed a higher hardness also due to their bigger shape, and this can be positively correlated with their DM content, which was the highest (Table 1). Nonetheless, it is important to underline that the hardness was determined on the products from the retail shops, where the exposure time was probably quite long at a temperature (room temperature, around 20 °C), which accelerated the ripening metabolism [33]. The total phenolic index (TPI) is known to be highly affected by temperature. In cooler conditions (<30 °C) with a pronounced deficit of radiation, concentrations of phenolic compounds were found to decrease (e.g., flavonols) and no further accumulation was observed [34]. The Rua cultivar samples showed higher concentrations of TPI in comparison to those of the Tahi cultivar; this is due to the reddish flesh of the former when mature. The TPI results were in accordance with the antioxidant capacity values being the highest in the Rua cultivar (Table 1). Among cultivars, it can be concluded that baby kiwifruit cultivar Rua shows the highest nutraceutical profile.

### 3.2. Chemical Characterization of the Kiwi Juices

Fresh juices were extracted from Tahi, Rua, and SunGold kiwifruits and subjected to chemical characterization (Table 2). SunGold juice exhibited higher sugar content and titratable acidity compared to the other varieties. Tahi juice displayed a higher pH and the lowest titratable acidity. Consistent with expectations, tartaric acid was not detected in any samples, while citric acid was found to be the major organic acid in line with previous findings [9]. The content of RAN was notable in all the juices, primarily represented by amino nitrogen. Inorganic nitrogen was not detected in SunGold juice, negligible in Tahi juice (2.7 mg/L), and significantly lower in Rua juice (26.0 mg/L) compared to amino nitrogen (173 mg/L). This characteristic suggests that these juices may be suitable for fermentation with minimal adjustment of nitrogen content to around 200 mg/L to ensure optimal yeast fermentative activity [35].

### 3.3. Aromatic Profile of the Kiwi Juices

Both free (Table 3) and glycosylated (Table 4) aroma compounds were determined in the investigated kiwi juices.

Thirteen free volatile compounds were detected belonging to acid, alcohol, and aldehyde groups, the latter being the less abundant in all samples (Table 3 and Table 4). Slight differences in the content of acids, particularly hexanoic acid and *trans*-2-hexenoic acid, were found among the juices. Notably, *trans*-2-hexenoic acid was recently suggested as a quality parameter due to its strong correlation with firmness and central red/blue color [36]. Tahi juice was the poorest in alcohols, in particular for *cis*-3-hexen-1-ol which conferred a grass note; this compound was found at a concentration lower than its perception threshold (400 μg/L [37]). This juice showed about a 2-fold higher concentration of 1-hexanol in comparison to Rua and SunGold juices; nonetheless, this alcohol, responsible for the resin, flower, and green notes, was revealed at a concentration lower than its perception threshold (1110 μg/L [38]). Rua juice was the richest in aldehydes (Table 3 and Table 4), with significant differences in *trans*-2-pentenal, although it was detected at a concentration below its perception threshold (1500 μg/L [39]). Additionally, nonanal was approximately 2-fold higher in Rua juice, conferring fat, citrus, and green notes, and it was detected at levels higher than its perception threshold in all samples, by approximately six, 10, and 16 times, respectively, for SunGold, Tahi, and Rua juices. Pentanal and hexanal were also detected in *Actinidia arguta* fruit, determined by olfactometry on the fruits [40].

**Table 3 foods-13-02380-t003:** Content of free aromatic compounds (expressed as μg/L) determined in kiwi juices from the three cultivars. Values are mean ± standard deviation of three replicates.

Compound	Perception Threshold (μg/L)	Descriptor	Tahi	Rua	SunGold
Acids					
Hexanoic acid	420 ^a^	Sweat	1078.5 ± 60.1 ^ab^	1022.5 ± 3.5 ^b^	1102.5 ± 24.7 ^a^
*trans*-2-Hexenoic acid	-	Must, fat	94.5 ± 7.8 ^a^	41.5 ± 4.9 ^c^	58.5 ± 4.9 ^b^
Total			1173.0 ± 67.9 ^a^	1064.0 ± 8.5 ^b^	1161.0 ± 29.7 ^a^
Alcohols					
1-Hexanol	1110 [38]	Resin, flower, green	185.0 ± 5.7 ^a^	99.0 ± 4.2 ^b^	90.5 ± 3.5 ^b^
2-Hexanol	-	Resin, flower, green	280.0 ± 31.1 ^b^	337.5 ± 24.7 ^a^	313.0 ± 2.8 ^ab^
*cis*-3-Hexen-1-ol	400 [37]	Grass	277.0 ± 17.0 ^c^	633.5 ± 30.4 ^b^	681.5 ± 4.9 ^a^
*trans*-2-Hexen-1-ol	-	Green, leaf, walnut	86.0 ± 5.7 ^a^	77.5 ± 10.6 ^a^	63.0 ± 2.8 ^b^
Homovanillic acid	-		34.5 ± 2.1 ^a^	19.0 ± 4.2 ^b^	30.5 ± 2.1 ^a^
Benzyl alcohol	10,000 [41]	Sweet, flower	33.5 ± 10.6 ^a^	48.5 ± 9.2 ^a^	39.0 ± 4.2 ^a^
2-Phenylethanol	10,000 [42]	Honey, spice, rose, lilac	28.0 ± 12.7 ^a^	37.5 ± 3.5 ^a^	25.0 ± 4.2 ^a^
Total			924.0 ± 84.9 ^b^	1252.5 ± 87.0 ^a^	1242.5 ± 24.7 ^a^
Aldehydes					
*trans*-2-Pentenal	1500 [39]	Fruit, tomato	237.0 ± 21.1 ^c^	522.5 ± 3.5 ^a^	420.0 ± 11.3 ^b^
Hexanal	25,000 [43]	Grass, tallow, fat	92.0 ± 5.7 ^a^	62.5 ± 17.7 ^b^	68.0 ± 9.9 ^ab^
*trans*-2-Hexenal	17 [41]	Green, leaf	100.5 ± 27.6 ^a^	90.5 ± 14.8 ^ab^	60.0 ± 14.1 ^b^
Nonanal	8 [42]	Fat, citrus, green	81.0 ± 5.7 ^b^	130.5 ± 14.8 ^a^	49.0 ± 1.4 ^c^
Total			510.5 ± 60.1 ^b^	806.0 ± 50.9 ^a^	597.0 ± 36.8 ^b^

Values in rows followed by the same superscript letter do not differ significantly (F-test, *p* ≤ 0.05).

The glycosylated compounds found in the kiwi juices belonged to the same groups as the free ones (Table 4). Eleven glycosylated aromas were detected in total. Glycosylated homovanillic acid, 2-hexanol, *trans*-2-hexenal, and nonanal were not detected in any of the juices, and glycosylated 4-methyl-benzoic acid was found only in Tahi juice. In particular, the latter showed the highest contents of glycosylated acids, especially hexanoic acid, alcohols, and aldehydes, while the Rua juice was the poorest in glycosylated acids and alcohols. Most of the found glycosylated compounds were previously detected in *A. arguta* fruit [40]. Nonetheless, Garcia et al. [40] found other bound aromatic compounds such as benzoic acid, cinnamic acid, and coniferyl alcohol, as well as norisoprenoid glycosides. These differences could be attributed to seasonal trends, area of origin, and the level of maturity, as certain volatile compounds can decrease during ripening [36].

**Table 4 foods-13-02380-t004:** Content of glycosylated aromatic compounds (expressed as μg/L) determined in kiwi juices from the three cultivars. Values are mean ± standard deviation of three replicates; n.d. = not detected.

Compound	Perception Threshold (μg/L)	Descriptor	Tahi	Rua	SunGold
Acids					
Hexanoic acid	420 ^a^	Sweat	1958.0 ± 89.1 ^a^	832.0 ± 28.3 ^c^	1259.0 ± 9.9 ^b^
*trans*-2-Hexenoic acid	-	Must, fat	110.5 ± 14.8 ^a^	n.d.	23.0 ± 2.8 ^b^
4-Methyl-benzoic acid	-	-	95.5 ± 14.8	n.d.	n.d.
Total			2164.5 ± 118.8 ^a^	832.0 ± 28.3 ^c^	1282.0 ± 12.7 ^b^
Alcohols					
1-Hexanol	1110 [38]	Resin, flower, green	685.5 ± 58.7 ^a^	375.0 ± 49.5 ^c^	483.0 ± 38.2 ^b^
2-Hexanol	-	Resin, flower, green	n.d.	n.d.	n.d.
*cis*-3-Hexen-1-ol	400 [37]	Grass	48.5 ± 4.9 ^a^	47.5 ± 3.5 ^a^	48.5 ± 9.2 ^a^
*trans*-2-Hexen-1-ol	-	Green, leaf, walnut	106.5 ± 7.8 ^a^	31.5 ± 4.9 ^c^	58.5 ± 4.9 ^b^
Benzyl alcohol	10,000 [41]	Sweet, flower	755.0 ± 23.3 ^a^	382.0 ± 24.0 ^c^	505.5 ± 74.2 ^b^
2-Phenylethanol	10,000 [42]	Honey, spice, rose, lilac	21.5 ± 0.7 ^c^	147.5 ± 3.5 ^b^	168.0 ± 2.8 ^a^
Total			1590.0 ± 95.5 ^a^	983.5 ± 85.6 ^c^	1263.5 ± 129.4 ^b^
Aldehydes					
*trans*-2-Pentenal	1500 [39]	Fuit, tomato	59.0 ± 12.7 ^a^	62.5 ± 17.7 ^a^	30.0 ± 7.1 ^b^
Hexanal	25,000 [43]	Grass, tallow, fat	128.5 ± 4.9 ^a^	44.5 ± 4.9 ^b^	30.5 ± 3.5 ^c^
Total			187.5 ± 17.7 ^a^	107.0 ± 22.6 ^b^	60.5 ± 10.6 ^c^

Values in rows followed by the same superscript letter do not differ significantly (F-test, *p* ≤ 0.05).

### 3.4. Fermentation Trials

According to Fracassetti et al. [17], fermentation processes were controlled by inoculating a pure culture of *T. delbrueckii* UMY196 as a starter yeast. The juices and kiwifruit-based drinks were analyzed at the beginning and at the end of the fermentations, respectively, to ensure a controlled process. Juices were evaluated for potential microbial contaminations, with no sporogenic or non-sporogenic contaminants detected. Subsequently, *T. delbrueckii* UMY196 was inoculated and the fermentation progress was monitored daily through the measurement of CO_2_ loss (gCO_2_/200 mL) (Figure 1). The end of the fermentation process was assessed when the weight difference compared to the previous day remained below 0.1 g/L for at least two consecutive days.

In all the trials, uniform yeast colonies were observed at the end of fermentation, and molecular identification confirmed the exclusive presence of the strain *T. delbrueckii* UMY196. It was observed that the fermentative performance of *T. delbrueckii* UMY196 varied depending on the fermented matrix. Notably, the strain exhibited a slower fermentative trend in SunGold fermentation, requiring a longer duration (25 days) for completion, whereas the fermentation time in Tahi and Rua juices was comparable (15 days each). However, a higher fermentative vigor was found within 5 days from the start of fermentation in all tests. Wei et al. [44] investigated the fermentation of fruit must prepared using green kiwifruit juice from *A. chinensis* species; depending on the type, variety, and pre-fermentation treatment of fruit, the application of non-*Saccharomyces* yeasts in fruit juice fermentation may lead to unpredictable modifications in the flavor profile. Similarly, Canonico et al. [45] observed divergent trends in two fermentation studies involving *T. delbrueckii*, suggesting that this variability could be attributed to differences in initial substrates and nutrient availabilities.

During the Tahi juice fermentation, an increase in yeast concentration from 5.7 × 10^6^ CFU/mL to 4.8 × 10^7^ CFU/mL was observed. Exponential growth started on day 1 and continued until day 7, after which it gradually decreased until day 17, when the residual sugar concentration reached 0.15 ± 0.01 g/L. Similarly, in Rua juice, *T. delbrueckii* adapted very quickly, with exponential growth occurring from day 1 to day 6 from the inoculum, followed by a gradual decrease until day 17, resulting in a final sugar content close to zero (0.20 ± 0.01 g/L) (Table 5). In SunGold juice, which exhibited the highest sugar concentration, the adaptation of *T. delbrueckii* was similar to other juices, with exponential growth occurring from day 1 to day 9 from the inoculum, followed by a decrease until day 11, resulting in a final sugar content of 0.13 ± 0.02 g/L (Table 5). This pattern is consistent with *T. delbrueckii*’s efficient fermentation of kiwifruit juice, as described by Fracassetti et al. (2019) [17].

Comparable glycerol levels were found in the samples, at 2.08 ± 0.21 g/L, 2.09 ± 0.21 g/L, and 1.85 ± 0.13 g/L for Tahi, Rua, and SunGold drinks, respectively (Table 5).

Furthermore, it was of paramount importance to analyze the type and quantity of organic acids, as these can significantly impact sensory and chemical characteristics such as pH, total acidity, and microbiological stability. [46]. Acidity is a peculiar characteristic of the obtained kiwifruit-based beverages. Specifically, the titratable acidity was 14.71 ± 0.58 g/L, 11.67 ± 0.47 g/L, and 16.75 ± 0.67 g/L (citric acid equivalents) for Tahi, Rua, and SunGold drinks, respectively (Table 5). These values were particularly relevant and can explain the high perception of acidity in the final drinks. The pH values (Table 5) did not correspond to the acidity as they were 3.32 ± 0.03, 3.46 ± 0.01, and 3.78 ± 0.09 for Tahi, Rua, and SunGold drinks, respectively. In particular, the pH of SunGold was the highest, and it had the highest acidity as well. Such differences can be explained by the content of malic acid, which was the lowest in the SunGold drink (1.77 ± 0.22 g/L). Moreover, the SunGold drink contained the highest concentration of acetic acid (0.62 ± 0.37 g/L), which was more than 2-fold higher than the Tahi and Rua drinks (0.20 ± 0.02 g/L and 0.28 ± 0.03 g/L for the Tahi and Rua drinks, respectively) (Table 5). Nonetheless, this concentration was not responsible for an unpleasant acetic acid note. The levels of succinic acid were comparable in all the kiwifruit-based drinks and tartaric and lactic acids were not detected in any of them. In particular, for lactic acid, this result indicates that the malolactic transformation did not occur. The obtained results corroborated previous studies conducted on *T. delbrueckii* strains. For instance, Ciani and Maccarelli [47] observed the production of low levels of succinic acid during fruit juice fermentations. Additionally, Wei et al. [44] used apple juice, among other substrates, and observed that the tested *T. delbrueckii* strain produced high levels of malic and citric acids, as well as lactic acid exclusively during the fermentation process.

#### 3.4.1. Aroma Profile

Thirty-six VOCs were detected, including eight acids, 13 alcohols, 13 esters, and two aldehydes (Table 6). The contribution of the fermented material was limited in the case of aldehydes, as only minor differences were observed. Similarly, the overall content of esters was comparable in the three kiwifruit drinks, except for ethyl hexanoate and ethyl octanoate, which were higher in the Tahi and SunGold samples, respectively. These aroma compounds are described as apple and peach (for ethyl hexanoate) and fruit (for ethyl octanoate), suggesting the potential fruity character of the kiwifruit drinks. Interestingly, both esters were present at concentrations much higher than their perception thresholds, up to 29- and 300-fold for ethyl hexanoate and ethyl octanoate, respectively. The Rua drink displayed major levels of both acids and alcohols. Among the latter, the content of phenylethanol was consistent in all samples, being approximately 3.5 to seven times higher than its perception threshold in the Thai and Rua drinks, respectively. This alcohol, described as honey, spice, rose, and lilac, could significantly impact the sensory characteristics of the products. Most of the volatiles detected in the green kiwifruit drink [17] were also found in the samples produced with Tahi, Rua, and SunGold varieties. These compounds included acids (isobutyric acid, hexanoic acid, 2-methylbutanoic acid, octanoic acid, decanoic acid, and decenoic acid), alcohols (1-hexanol, 2-hexanol, 3-ethoxy-1-propanol, *cis*-3-hexen-1-ol, isoamyl alcohol, 2,3-butanediol, 2-phenylethanol, and *p*-tyrosol), esters (isoamyl acetate, ethyl acetate, ethyl hexanoate, ethyl octanoate, phenylethyl acetate, diethyl malate, and ethyl hydrogen succinate), and an aldehyde (phenylethanal). Moreover, benzyl alcohol was detected only in its glycosylated form in the green kiwifruit drink [17]. Interestingly, phenylethanol and *p*-tyrosol were much more prevalent in the kiwifruit drinks produced in this work. Although *p*-tyrosol, a phenolic alcohol, has a negligible effect on the sensory characteristics of the kiwi-based drinks, it can prevent the oxidative degradation of unsaturated fatty acids and amino acids [48]. Butyric acid and its esters were found in wines from green kiwifruit [49], fermented with *Saccharomyces* and non-*Saccharomyces* yeast [49] and produced with different chaptalization strategies [50]. Additionally, two lactones, γ-butyrolactone and α-methyl-γ-butyrolactone, mainly associated with wood aging, were found in the kiwifruit drinks; the lactones were also detected in other kiwifruit wines [51].

#### 3.4.2. Sensory Analysis

The descriptive quantitative profile of the kiwifruit-based drinks was evaluated by an expert panel (Figure 2).

A high perception of acidity was found, which can be attributed to the relevant content of citric acid (Table 5). The selected descriptors aligned well with the detected VOCs. Specifically, the descriptors of fruity, passion fruit, honey and floral, apple and peach, and citrus notes were associated with phenylethanal, 2-phenylethanol, ethyl octanoate, ethyl hexanoate, and 1-hexanol, respectively, and these compounds exhibited the highest odor activity values (OAVs) (Section 3.4.2). Notably, differences were observed in the olfactory perception of the passion fruit descriptor in the Tahi drink and in the aftertaste perception of fruity and apple notes in the SunGold drink. Even if the Rua drink had the highest overall content of aroma compounds (Table 6), the scores for the selected descriptors were lower in comparison to the Tahi and SunGold drinks. These findings could be attributed to the higher acid content in the Rua drink, which may attenuate and alter the perception of its fruity and floral characteristics.

To better clarify the relationship between the volatile compounds, which were significantly different in the three kiwi-based drinks, and sensory attributes, Principal Component Analysis (PCA) was conducted (Figure 3). The results showed that the first two components were significant, explaining 92% of the total variance, with 58% explained by P1 and 34% by P2. The Tahi drink was mainly characterized by volatile compounds described as rose and honey (phenylethyl acetate) and sweet (ethyl isobutyrate). The Rua drink exhibited wine and fruity notes (ethyl hydrogen succinate), as well as caramel and sweet notes (γ-butyrolactone). The SunGold drink was highlighted by its fruity notes, attributed to ethyl octanoate and hexyl acetate.

## 4. Conclusions

The comprehensive evaluation of the physicochemical and nutraceutical characteristics of different kiwifruit varieties, as well as the chemical composition and fermentation potential of their juices, provides valuable insights into their quality and possible applications. This study demonstrated the suitability of kiwifruit juices from various species for producing alcoholic beverages using a non-*Saccharomyces* yeast strain. The resulting sensory characteristics of the products depend on the quality of the fruits, the chemical and aromatic composition of the juice, and fermentation by *T. delbrueckii*, which enriches the sensory complexity of kiwi-based drinks. Indeed, the analysis revealed significant differences in the volatile compound profiles and sensory attributes of the drinks, offering the potential for product differentiation in the market. Among the three varieties tested, the most promising seems to be the Tahi if the desired characteristics include the passion fruit characteristic. Nonetheless, pleasant drinks can also be obtained with the SunGold due to its apple note.

These findings provide valuable insights into the physicochemical and sensory characteristics of various kiwifruit varieties, highlighting their potential applications in the food and beverage industry with a sustainable approach to reducing food waste of fruits rich in bioactive compounds. The evaluation of kiwifruit varieties and their juice fermentation potential facilitates the creation of unique alcoholic beverages with distinct sensory profiles, thereby enhancing product differentiation for targeted consumer segments. This promotes sustainable practices through the utilization of fruit waste and supports the development of functional beverages rich in bioactive compounds. The growing consumer interest in health-conscious, value-added products is expected to drive market growth for kiwi-based beverages. Furthermore, collaborative efforts among researchers, industry stakeholders, and agricultural producers could further amplify sustainable and economic benefits.

## Figures and Tables

**Figure 1 foods-13-02380-f001:**
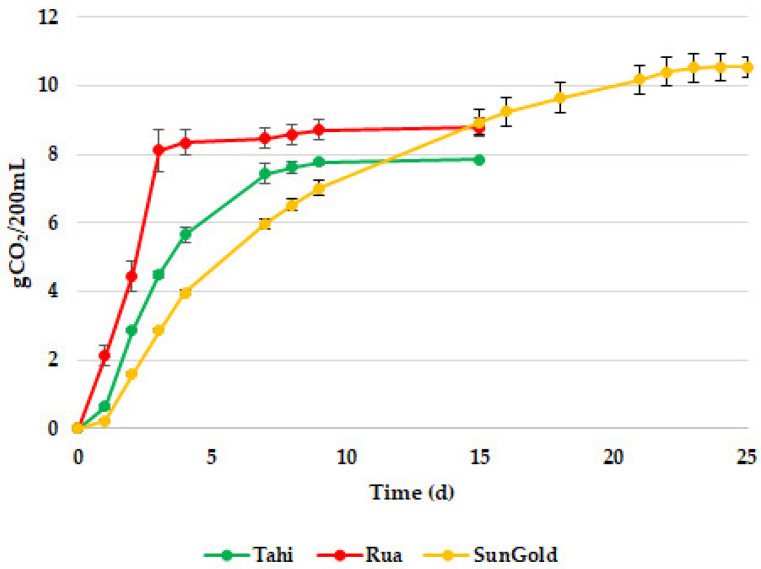
Fermentation trends of kiwifruit juices inoculated with *T. delbrueckii* UMY196 and monitored through the measurement of CO_2_ loss (gCO_2_/200 mL). The reported data are an average of three replicates and the bars indicate the standard deviations.

**Figure 2 foods-13-02380-f002:**
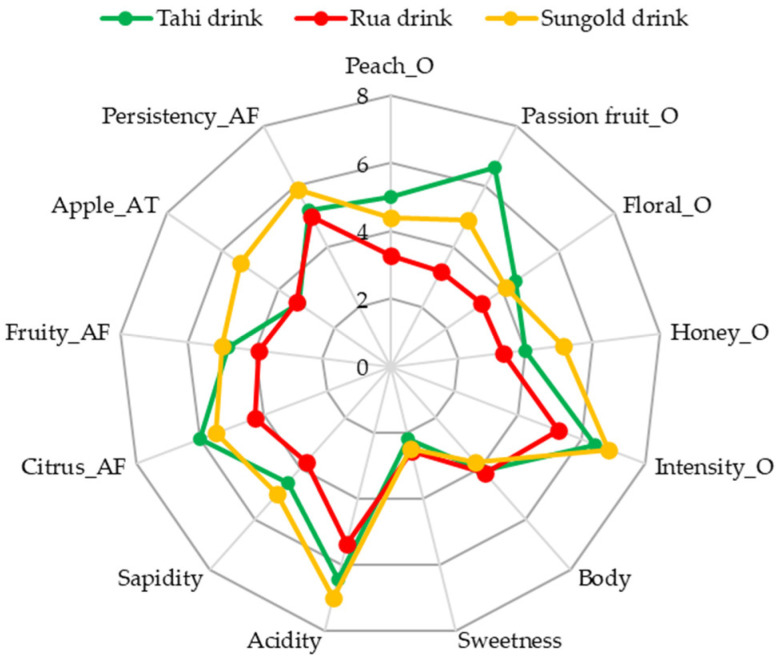
Descriptive quantitative profile of drink samples from the three kiwifruit cultivars fermented with *Torulaspora delbrueckii* UMY196.

**Figure 3 foods-13-02380-f003:**
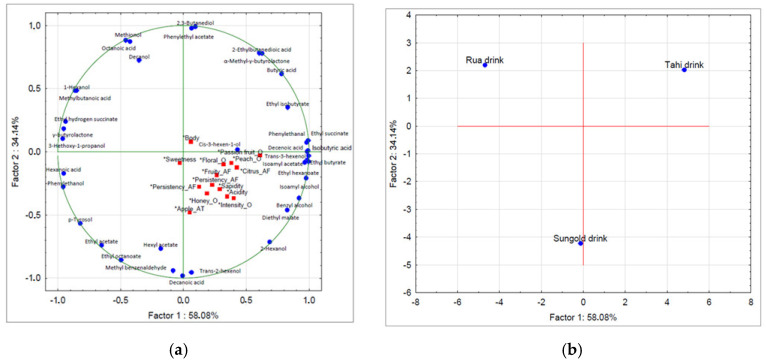
Projection of the (**a**) scores and (**b**) loading on the factor-plane obtained for aroma compounds and sensory scores of the investigated kiwifruit drinks. The volatile variables (in blue) were set as active variables and sensory variables (in red) as supplementary variables. Legend: O, olfactory; AT, aftertaste.

**Table 1 foods-13-02380-t001:** Results of physiochemical and nutraceutical measurements of the three kiwifruit cultivars. Values are mean ± standard deviation of twelve replicates. TSS: Total soluble solids.

Fresh Kiwifruit cv	TSS (°Brix)	Acidity (%)	Hardness (N)	Dry Matter (%)	Total Phenolic Index (mg/100 g Gallic Acid Equivalents)	Antioxidant Capacity (mmol Fe^2+^/kg)
Tahi	11.7 ± 0.9 ^ab^	2.0 ± 0.2 ^ab^	4.98 ± 0.8 ^b^	18.5 ± 0.9 ^ab^	187.09 ± 13.89 ^ab^	45.09 ± 1.99 ^b^
Rua	10.3 ± 1.1 ^b^	2.2 ± 0.3 ^a^	3.57 ± 0.5 ^b^	19.8 ± 1.3 ^b^	209.36 ± 12.36 ^a^	57.98 ± 2.98 ^a^
SunGold	12.9 ± 0.6 ^a^	1.5 ± 0.2 ^b^	6.45 ± 0.9 ^a^	22.4 ± 0.7 ^a^	168.25 ± 19.69 ^b^	29.87 ± 1.89 ^c^

Values in columns followed by the same superscript letter do not differ significantly according to the F-test (*p* ≤ 0.05).

**Table 2 foods-13-02380-t002:** Chemical characterization of the kiwifruit juices from the three cultivars. Values are mean ± standard deviation of three replicates; n.d. = not detected.

	Tahi	Rua	SunGold
Sugars (g/L)	102.63 ± 7.49 ^b^	140.80 ± 7.42 ^a^	141.47 ± 0.40 ^a^
Readily assimilable nitrogen (mg N/L)	121 ± 4.84 ^b^	199 ± 7.96 ^a^	204 ± 9.18 ^a^
pH	3.38 ± 0.01 ^c^	3.64 ± 0.02 ^a^	3.60 ± 0.02 ^b^
Titratable Acidity (g citric acid/L)	11.45 ± 0.34 ^c^	12.59 ± 0.37 ^b^	14.19 ± 0.38 ^a^
Malic Acid (g/L)	1.92 ± 0.04 ^a^	1.74 ± 0.04 ^b^	1.19 ± 0.02 ^c^
Citric Acid (g/L)	7.94 ± 0.16 ^a^	5.26 ± 0.11 ^c^	6.18 ± 0.12 ^b^
Succinic Acid (g/L)	0.46 ± 0.02 ^a^	0.31 ± 0.01 ^b^	n.d.

Values in rows followed by the same superscript letter do not differ significantly (F-test, *p* ≤ 0.05).

**Table 5 foods-13-02380-t005:** Chemical characterization of the drink samples from the three kiwifruit cultivars at the end of the fermentation processes. Values are mean ± standard deviation of three replicates; n.d. = not detected.

	Tahi	Rua	SunGold
Sugars (g/L)	0.15 ± 0.03 ^b^	0.20 ± 0.01 ^a^	0.13 ± 0.00 ^b^
pH	3.32 ± 0.03 ^c^	3.46 ± 0.01 ^b^	3.78 ± 0.09 ^a^
Total Acidity (g citric acid/L)	14.71 ± 0.58 ^b^	11.67 ± 0.47 ^c^	16.75 ± 0.67 ^a^
Ethanol (g/L)	6.46 ± 1.47 ^a^	7.90 ± 1.04 ^a^	8.85 ± 1.06 ^a^
Glycerol (g/L)	2.08 ± 0.21 ^a^	2.09 ± 0.21 ^a^	1.85 ± 0.13 ^a^
Malic Acid (g/L)	2.19 ± 0.29 ^a^	2.31 ± 0.27 ^ab^	1.77 ± 0.22 ^b^
Lactic Acid (g/L)	n.d.	n.d.	n.d.
Acetic Acid (g/L)	0.21 ± 0.03 ^a^	0.24 ± 0.06 ^a^	0.63 ± 0.37 ^a^
Citric Acid (g/L)	8.96 ± 0.50 ^a^	6.66 ± 0.75 ^b^	8.86 ± 1.30 ^a^
Succinic Acid (g/L)	0.91 ± 0.11 ^a^	0.90 ± 0.02 ^a^	0.94 ± 0.13 ^a^

Values in rows followed by the same superscript letter do not differ significantly (F-test, *p* ≤ 0.05).

**Table 6 foods-13-02380-t006:** Content of free aromatic compounds (expressed as μg/L) determined in the drink samples from the three kiwi cultivars at the end of the fermentation processes. Values are mean ± standard deviation of three replicates; n.d. = not detected.

Compound	Perception Threshold (μg/L)	Descriptor	Tahi	Rua	SunGold
Acids					
Isobutyric acid	2300 [52]	Rancid, butter, cheese	512.5 ± 17.7 ^a^	321.5 ± 27.6 ^c^	413.5 ± 2.1 ^b^
Hexanoic acid	420 [52]	Sweat	2092.0 ± 48.1 ^c^	3138.5 ± 116.7 ^a^	2773.5 ± 111.0 ^b^
Butyric acid	240 [41]	Sweat, rancid, cheese	448.0 ± 5.7 ^a^	333.5 ± 12.0 ^b^	310.0 ± 7.1 ^c^
2-Methylbutanoic acid	33 [42]	Cheese, sweat	157.0 ± 4.2 ^a^	352.0 ± 9.9 ^b^	161.0 ± 2.8 ^a^
2-Ethylbutanedioic acid	-	-	385.0 ± 9.9 ^a^	234.5 ± 3.5 ^b^	143.5 ± 10.6 ^c^
Octanoic acid	500 [52]	Cheese, sweat	3232.0 ± 5.7 ^b^	4036.5 ± 21.9 ^a^	2326.0 ± 26.9 ^c^
Decanoic acid	1000 [42]	Rancid, fat	410.5 ± 14.8 ^b^	412.5 ± 12.0 ^b^	498.5 ± 4.9 ^a^
Decenoic acid	2 [42]	Fat	249.5 ± 2.1 ^a^	202.5 ± 3.5 ^c^	225.5 ± 7.8 ^b^
Total			7486.5 ± 108.2 ^b^	9031.5 ± 207.2 ^a^	6851.5 ± 173.2 ^c^
Alcohols					
1-Hexanol	1110 [38]	Resin, flower, green	1120.0 ± 43.8 ^a^	2325.0 ± 35.4 ^b^	1138.0 ± 159.8 ^a^
2-Hexanol	-	Resin, flower, green	72.5 ± 3.5 ^b^	21.5 ± 4.9 ^c^	92.5 ± 10.6 ^a^
3-Ethoxy-1-propanol	-	Fruit	n.d.	23.5 ± 2.1.	9.5 ± 3.5.
*cis*-3-Hexen-1-ol	400 [37]	Grass	27.5 ± 6.4 ^a^	23.5 ± 6.4 ^a^	25.0 ± 1.4 ^a^
*trans*-2-Hexenol	-	Green, leaf, walnut	26.5 ± 19.1 ^b^	22.5 ± 3.5 ^b^	92.5 ± 10.6 ^a^
*trans*-3-Hexenol		Moss, fresh	621.0 ± 2.8 ^a^	413.0 ± 2.8 ^c^	522.0 ± 9.9 ^b^
Decanol		Fat	42.5 ± 3.5 ^ab^	47.5 ± 3.5 ^a^	37.0 ± 2.8 ^b^
Isoamyl alcohol	30,000 [42]	Spirit, alcoholic	8003.0 ± 25.5 ^a^	5140.5 ± 29.0 ^c^	7082.5 ± 60.1 ^b^
2,3-Butanediol	-	Fruit, onion	1810.0 ± 21.2 ^a^	1762.5 ± 16.3 ^b^	1293.5 ± 12.0 ^c^
2-Phenylethanol	10,000 [42]	Honey, spice, rose, lilac	34,999 ± 9 ^c^	70,193 ± 16 ^a^	61,565 ± 622 ^b^
Benzyl alcohol	10,000 [41]	Sweet, flower	92.5 ± 3.5 ^a^	59.0 ± 5.7 ^b^	87.0 ± 2.8 ^a^
Methionol	-	Sweet, potato	27± 13.4 ^c^	349.0 ± 9.9 ^a^	187.0 ± 18.4 ^b^
*p*-Tyrosol	-	-	1240.5 ± 6.4 ^c^	2018.5 ± 4.9 ^b^	2097.0 ± 11.3 ^a^
Total			47,090 ± 152 ^c^	82,400 ± 136 ^a^	72,132 ± 152 ^b^
Aldehydes					
Phenylethanal	1 [37]	Honey, sweet, hawthorn	20.5 ± 0.7 ^a^	11.0 ± 1.4 ^c^	15.0 ± 1.4 ^b^
Methyl benzenaldehyde	-	-	108.0 ± 4.2 ^b^	113.5 ± 16.3 ^b^	155.0 ± 4.2 ^a^
Total			128.5 ± 4.9 ^b^	124.5 ± 17.7 ^b^	170.0 ± 5.7 ^a^
Esters					
Isoamyl acetate	12,270 [42]	Banana	413.0 ± 11.3 ^a^	237.0 ± 21.1 ^c^	333.5 ± 26.2 ^b^
Ethyl acetate	7500 [42]	Pineapple	n.d.	52.5 ± 3.5 ^b^	77.5 ± 10.6 ^a^
Ethyl butyrate	1 [41]	Apple	110.0 ± 14.1 ^a^	55.0 ± 7.1 ^c^	86.0 ± 1.4 ^b^
Ethyl isobutyrate	15 [53]	Sweet, rubber	47.5 ± 10.6 ^a^	22.5 ± 3.5 ^b^	25.0 ± 7.1 ^b^
Ethyl hexanoate	14 [54]	Apple, peach	406.0 ± 8.5 ^a^	188.0 ± 19.8 ^c^	310.0 ± 7.1 ^b^
Ethyl octanoate	2 [55]	Fruit, fat	412.5 ± 17.7 ^c^	510.0 ± 7.1 ^b^	606.0 ± 8.5 ^a^
Ethyl succinate			655.5 ± 7.8 ^a^	439.5 ± 12.0 ^c^	564.5 ± 13.4 ^b^
Phenylethyl acetate	250 [42]	Rose, honey, tobacco	250.0 ± 4.2 ^a^	202.0 ± 4.2 ^a^	147.5 ± 6.4 ^b^
Diethyl malate	-	Brown sugar, sweet	129.0 ± 1.4 ^a^	109.5 ± 6.4 ^b^	128.5 ± 4.9 ^a^
Ethyl hydrogen succinate	-	Wine, fruit	5557.5 ± 10.6 ^c^	6926.5 ± 149.2 ^a^	5938.5 ± 160.5 ^b^
Hexyl acetate	2 [41]	Fruit	140.0 ± 14.1 ^b^	165.0 ± 49.5 ^ab^	225.0 ± 34.5 ^a^
γ-Butyrolactone	-	Caramel, sweet	30.0 ± 2.8 ^c^	89.5 ± 4.9 ^a^	50.0 ± 7.1 ^b^
α-Methyl-γ-butyrolactone	-	Woody	137.5 ± 4.9 ^a^	104.0 ± 5.7 ^b^	81.5 ± 4.9 ^c^
Total			8253.5 ± 108.2 ^b^	9155.0 ± 294.2 ^a^	8573.5 ± 293.4 ^b^

Values in rows followed by the same superscript letter do not differ significantly (F-test, *p* ≤ 0.05).

## Data Availability

The original contributions presented in the study are included in the article, further inquiries can be directed to the corresponding author.

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
