# Peer review of "From Fruit to Beverage: Investigating Actinidia Species for Characteristics and Potential in Alcoholic Drink Production"

_foods, 2024, doi:10.3390/foods13152380_

Round 1

Reviewer 1 Report

Comments and Suggestions for Authors

The article titled "From fruit to juice: exploring the Actinidia species"  investigated different ciwi fruit cultivars physio-chemically to find  the suitability of these juices for alcoholic beverage production using non-Saccharomyces strains. It is a well-written article and can be further considered after minor revision. Authors should more focus on the differences of the different cultivars in the results and discussion section and make a discussion and conclusion out of it. My specific comments are given in the pdf file. 

Thank you

Comments on the Quality of English Language

Minor improvements are needed. 

Author Response

Thank you very much for taking the time to review this manuscript. Please find the detailed responses below and the corresponding revisions/corrections highlighted changes in the re-submitted files.

Point-by-point response to Comments and Suggestions for Authors

Comments 1: Incomplete title. Suggest to improve it as: From fruit to beverage: investigating Actinidia species for characteristics and potential in alcoholic drink production

Response 1: Thank you for pointing this out. We agree with this comment. Therefore, we have changed the title of the paper as you suggested: “From fruit to beverage: investigating Actinidia species for characteristics and potential in alcoholic drink production”.

Comments 2: Please revise the abstract thoroughly: Methods are not clearly described and treatments are vague. Results are not properly reported. Suggest to divide your results based on the experiments you have conducted and clearly mention the significant differences among the treatments. In the end of abstract, add a conclusion

Response 2: As the reviewer suggested, the abstract was modified resuming the general part and including more details about the results (lines 23-33).

Comments 3: Citric acid.

Response 3: Done.

Comments 4: Full name please.

Response 4: Done.

Comments 5: What about SunGold?

Response 5: Thank you for pointing this out. We agree with this comment and we changed the text as follow: “The outcome of this work may facilitate the reduction of waste and the market expansion of these kiwi species through the implementation of alternative processing methods” (lines 101-103).

Comments 6: Please mention how much juice was obtained from each fruit.

Response 6: As the reviewer suggested, the juice extraction yields were included: “…obtaining a juice extraction yield up to 70%, specifically 70%, 32% and 65% for SunGold, Rua and Tahi, respectively.” (lines 114-116).

Comments 7: What is the difference between 2.2 and 2.3; both of which are analyzing physiochemical of juice

Response 7: Probably it was a misunderstanding. In Section 2.2. we described the analyses performed on the fruit pulp, while in the Section 2.3 we described the analyses performed on kiwifruit juices obtained with the extractor.

Comments 8: Please clearly mention fruit or juice. If juice, you can report at 2.3 otherwise, why you didn't use pulp for the analysis.

Response 8: We agree with this comment; it was a refuse. The titration was made on extracted pulp of kiwi with the seed elimination. Therefore, we added this sentence “Pulp fruit from each sample was extracted and the titratable acidity (TA) was measured with an automatic titrator; results were expressed as % (w/v) acid citric.” (lines 122-124).  

Comments 9: What about physical characterization?

Response 9: It was a preliminary evaluation of these juice and we have focused our attention on chemical aspects; we modified the title of the section as follows “Chemical and Nutraceutical Evaluation of Kiwifruit” (line 119).

Comments 10: Kiwi juice of drink?

Response 10: In this context we were speaking about the product of the fermentation, so it was correct talk about drink. We decided to refer to the product as drink, while the raw material as juice.

Comments 11: Please use either juice or drink.

Response 11: We decided to refer to the product as drink, while the raw material as juice.

Comments 12: Please focus more on your results and differences between cultivars. Here, I only see some explanations about hardness. not about the results and findings.

Response 12: Thank you for your comment. We have improved this part as follows: “Among varieties the SunGold is the sweater with also the lowest acidity (%) value while the cv Rua is the baby kiwi with the lowest TSS content.” (lines 265-266); “With regard to the dry matter (DM) measurements, minimal differences were observed in the baby kiwi samples, whereas variations were evident in the SunGold with the highest level (22%).” (lines 273-275); “In particular, Oh et al. [14] indicate that for the hardy kiwifruit, the development of DM, starch and sugars occurs independently of environmental conditions in pre-harvest. In our case all samples with Nergi® brand are from the same cultivation area (Cuneo province, Italy).” (lines 278-281); “Hardness values found in this work were generally higher for all the kiwifruit varieties considered than those found as outcomes [33]. In any case the SunGold fruits showed a higher hardness, due also their bigger shape, and this can be positive correlated with the highest DM content (Table 1).” (lines 294-297); “Among cultivars can be concluded that baby kiwifruit cultivar Rua show the highest nutraceutical profile.” (lines 307-308).

Comments 13: Please insert full name in caption.

Response 13: Amended.

Comments 14: There is not a single sentence about the differences between the cultivars. You used 3 cultivars in this study. Please mention which one is more promising.

Response 14: As the reviewer suggested, we proposed the most promising cultivars in lines 580-583: “Among the three varieties tasted, the most promising seems to be the Tahi if the desired characteristics include the passion fruit character. Nonetheless, pleasant drinks can be also obtained with the SunGold due to its apple note.”

Comments 15: Please write about the future direction as well.

Response 15: As the reviewer suggested, the future perspectives were reported in lines 584-595: “These findings provide valuable insights into the physicochemical and sensory characteristics of various kiwifruit varieties, highlighting their potential applications in the food and beverage industry with a sustainable approach to reduce food waste of fruits rich in bioactive compounds. The evaluation of kiwifruit varieties and their juice fermentation potential facilitates the creation of unique alcoholic beverages with distinct sensory profiles, thereby enhancing product differentiation for targeted consumer segments. This promotes sustainable practices through the utilization of fruit waste and supports the development of functional beverages rich in bioactive com-pounds. The growing consumer interest in health-conscious, value-added products is expected to drive market growth for kiwi-based beverages. Furthermore, collaborative efforts among researchers, industry stakeholders, and agricultural producers could further amplify sustainable and economic benefits.”

Reviewer 2 Report

Comments and Suggestions for Authors

The manuscript entitled "From Fruit to Juice: Exploring the Actinidia Species" is currently unsuitable for publication. The study appears to be based on local work, which should be repeated over two consecutive years to ensure reliability.

The timing of the fruit harvest, based on TSS, indicates that the fruits were harvested too late, leading to over-ripening. This over-ripening likely affected other parameters. Additionally, the observed sugar levels are insufficient for yeast survival, hindering the fermentation process.

Throughout the manuscript, the term "kiwi" should be replaced with "kiwifruit".

The abstract should be revised to include more detailed results, moving general information to the Introduction section.

Author Response

Thank you very much for taking the time to review this manuscript. Please find the detailed responses below and the corresponding revisions/corrections highlighted changes in the re-submitted files.

Comments 1: The manuscript entitled "From Fruit to Juice: Exploring the Actinidia Species" is currently unsuitable for publication. The study appears to be based on local work, which should be repeated over two consecutive years to ensure reliability.

Response 1: The study aimed to evaluate the chemical composition of kiwifruit juices obtained from different varieties (SunGold, Rua and Tahi) in order to increase the knowledge related to these fruits. Moreover, the suitability of juices for the production of low-alcoholic beverages. We agree that the chemical characterization should cover two vintages since the chemical parameters are affected by the season. However, the major results showed the peculiar characteristics of the different kiwifruit varieties under study as well as the feasibility of producing fermented beverages from these fruits. This aspect is of interest for the producers because of the short shelf life of baby kiwis in particular. As a consequence, a reliable transformation process can allow to limit the food loss and waste with a possible positive impact on the food system. For all these reasons, we believe that such investigation can represent an advantage as well as an interesting tool for supporting the kiwifruit growers as well as the distribution market giving a second life to the excess of production, unperfect and unsold fruits.   

Comments 2: The timing of the fruit harvest, based on TSS, indicates that the fruits were harvested too late, leading to over-ripening. This over-ripening likely affected other parameters. Additionally, the observed sugar levels are insufficient for yeast survival, hindering the fermentation process.

Response 2: We agree with this comment. The fruits are at the late ripening time because the idea is to use them in a circular economy, reducing waste of overripening fruit or discarded fruits from the storage management in the view of kiwi juice valorization.

The sugar content of the kiwifruit juices, as shown in Table 2, was sufficient for yeast survival. Specifically, the sugar concentrations in the juices were 102.63 g/L, 140.80 g/L, and 141.47 g/L for the Tahi, Rua, and SunGold cultivars, respectively. In laboratory experiments, synthetic media like Yeast Nitrogen Base typically contain only 2% glucose. Additionally, the fermentations were not limited, as indicated by the final concentration of sugars being lower than 0.2 g/L (Table 5).

Comments 3: Throughout the manuscript, the term "kiwi" should be replaced with "kiwifruit".

Response 3: Amended in the entire text.

Comments 4: The abstract should be revised to include more detailed results, moving general information to the Introduction section.

Response 4: As the reviewer suggested, the abstract was modified resuming the general part and including more details about the results (lines 23-33).

Round 2

Reviewer 2 Report

Comments and Suggestions for Authors

Accept in present form